# Using contact network dynamics to implement efficient interventions against pathogen spread in hospital settings: A modelling study

**Quentin J. Leclerc**[1,2,3☺*], **Audrey Duval**[1,2,3☺¤], **Didier Guillemot**[1,2,4], **Lulla Opatowski**[1,2‡], **Laura Temime**[3,5‡]

**1** Institut Pasteur, Université Paris Cité, Epidemiology and Modelling of Bacterial Escape to Antimicrobials (EMEA), Paris, France, **2** INSERM, Université Paris-Saclay, Université de Versailles St-Quentin-en-Yvelines, Team Echappement aux Anti-infectieux et Pharmacoépidémiologie U1018, CESP, Versailles, France, **3** Laboratoire Modélisation, Epidémiologie et Surveillance des Risques Sanitaires, Conservatoire National des Arts et Métiers, Paris, France, **4** AP-HP, Paris Saclay, Department of Public Health, Medical Information, Clinical Research, Garches, France, **5** Institut Pasteur, Conservatoire National des Arts et Métiers, Unité PACRI, Paris, France

☺ These authors contributed equally to this work.
¤ Current address: Imagine Institute, Data Science Platform, INSERM UMR 1163, Université de Paris, Paris, France
‡ LO and LT also contributed equally to this work.
* quentin.leclerc@pasteur.fr

**Data Availability Statement:** This study makes use of data previously collected as part of the i-Bird study. Full details are available at: https://doi.org/10.1371/journal.pcbi.1004170. The relevant data are included in the repository which contains the

## Abstract

### Background

Long-term care facilities (LTCFs) are hotspots for pathogen transmission. Infection control interventions are essential, but the high density and heterogeneity of interindividual contacts within LTCF may hinder their efficacy. Here, we explore how the patient–staff contact structure may inform effective intervention implementation.

### Methods and findings

Using an individual-based model (IBM), we reproduced methicillin-resistant *Staphylococcus aureus* colonisation transmission dynamics over a detailed contact network recorded within a French LTCF of 327 patients and 263 staff over 3 months. Simulated baseline cumulative colonisation incidence was 21 patients (prediction interval: 11, 31) and 35 staff (prediction interval: 19, 54). We examined the potential impact of 3 types of interventions against transmission (reallocation reducing the number of unique contacts per staff, reinforced contact precautions, and hypothetical vaccination protecting against acquisition), targeted towards specific populations. All 3 interventions were effective when applied to all nurses or healthcare assistants (median reduction in MRSA colonisation incidence up to 35%), but the benefit did not exceed 8% when targeting any other single staff category.

We identified "supercontactor" individuals with most contacts ("frequency-based," over-represented among nurses, porters, and rehabilitation staff) or with the longest cumulative

model code: https://gitlab.pasteur.fr/qleclerc/ctcmodeler.

**Funding:** QJL, DG, LO and LT received funding from the Innovative Medicines Initiative 2 Joint Undertaking under grant agreement No 101034420 (PrIMAVeRa, https://www.imi.europa.eu/). This Joint Undertaking receives support from the European Union's Horizon 2020 research and innovation programme and EFPIA. DG received funding from the National Clinical Research Program and the Investissement d'Avenir program, Laboratoire d'Excellence "Integrative Biology of Emerging Infectious Diseases" (ANR-10-LABX-62-IBEID). The funders had no role in study design, data collection and analysis, decision to publish, or preparation of the manuscript.

**Competing interests:** LO reports grants from Pfizer outside the submitted work. The authors have declared that no other competing interests exist.

**Abbreviations:** COVID-19, Coronavirus Disease 2019; CPI, close proximity interaction; HAI, healthcare-associated infection; HCW, healthcare worker; IBM, individual-based model; LTCF, long-term care facility; MRSA, methicillin-resistant *Staphylococcus aureus*; PVS, persistent vegetative state.

time spent in contact ("duration-based," overrepresented among healthcare assistants and patients in elderly care or persistent vegetative state (PVS)). Targeting supercontactors enhanced interventions against pathogen spread in the LTCF. With contact precautions, targeting frequency-based staff supercontactors led to the highest incidence reduction (20%, 95% CI: 19, 21). Vaccinating a mix of frequency- and duration-based staff supercontactors led to a higher reduction (23%, 95% CI: 22, 24) than all other approaches. Although based on data from a single LTCF, when varying epidemiological parameters to extend to other pathogens, our results suggest that targeting supercontactors is always the most effective strategy, indicating this approach could be applied to prevent transmission of other nosocomial pathogens.

## Conclusions

By characterising the contact structure in hospital settings and identifying the categories of staff and patients more likely to be supercontactors, with either more or longer contacts than others, interventions against nosocomial spread could be more effective. We find that the most efficient implementation strategy depends on the intervention (reallocation, contact precautions, vaccination) and target population (staff, patients, supercontactors). Importantly, both staff and patients may be supercontactors, highlighting the importance of including patients in measures to prevent pathogen transmission in LTCF.

## Author summary

### Why was this study done?

- Infection control in healthcare centres such as long-term facilities (LCTFs) is challenging due to high-density and varied contact patterns among individuals.

- Understanding the contact structure between and within patients and healthcare workers and its impact on transmission could offer new perspectives to improve the effectiveness of infection control interventions.

### What did the researchers do and find?

- We developed an individual-based model (IBM) of methicillin-resistant *Staphylococcus aureus* (MRSA) colonisation dynamics, informed by a detailed contact network and epidemiological data collected within an LTCF over 3 months.

- We used simulations to evaluate 3 intervention types: reallocation assigning patients to a given staff member of each category throughout their entire stay, reinforced contact precautions, and hypothetical vaccination reducing acquisition risk.

- We identified "supercontactors" with the most contacts (frequency-based, overrepresented among nurses, porters, and rehabilitation staff) or longest time spent in contact (duration-based, overrepresented among healthcare assistants and patients in elderly care or persistent vegetative state (PVS)).

- Simulations revealed that targeting supercontactors enhanced intervention efficacy, with contact precautions reducing MRSA colonisation incidence by up to 20% (95% CI: 19, 21) and vaccination by up to 23% (95% CI: 22, 24).

### What do these findings mean?

- Infection control measures can be optimised by identifying and targeting supercontactors among staff and patients.

- Both staff and patients can be supercontactors, highlighting the need to include both groups in infection prevention strategies.

- The most effective strategy depends on both the intervention (reallocation, contact precautions, vaccination) and the subpopulation targeted (staff, patients, supercontactors).

- Although our quantitative conclusions are informed by data from a single LTCF by varying epidemiological parameters to explore other pathogens, we suggest that interventions targeting supercontactors could be applied to prevent the spread of other nosocomial pathogens.

## Introduction

Healthcare-associated infections (HAIs) are a major threat worldwide, with more than 4 million infections occurring each year in Europe [1]. The recent Coronavirus Disease 2019 (COVID-19) pandemic has underlined the high risk of pathogen dissemination in health care settings, similarly to what was previously reported for other coronaviruses, seasonal influenza, or Ebola epidemics [2,3]. Bacterial nosocomial outbreaks are also frequently described, becoming more and more difficult to control with the rise of multidrug resistance [4]. In addition to significantly impacting the morbidity and mortality of hospitalised patients and potentially healthcare workers (HCWs), HAI generate additional costs due to longer hospital stays or additional expensive therapeutics, as well as legal consequences for practitioners and healthcare settings in case of patient lawsuits.

Methicillin-resistant *Staphylococcus aureus* (MRSA) is an important cause of such HAI, as these infections most often affect individuals in a weakened immunological state, such as hospitalised patients [5]. Crucially, MRSA colonisation is a risk factor for infection, since individuals are more likely to be infected by an *S. aureus* strain they are carrying [6]. Consequently, it is essential to understand how individuals become colonised by MRSA in healthcare settings and to control the acquisition risk.

To limit pathogen dissemination through human cross-transmission in healthcare settings, a range of measures can be implemented, mostly based on improving contact precautions, such as patient isolation, hand-washing, wearing of gloves or masks. Vaccines to reduce the risk of pathogen colonisation also represent ongoing research and development topics, although none are commercially available and there have only been limited attempts to evaluate their impact in healthcare settings thus far [7]. However, the high density of human contacts involving HCWs, patients, and visitors, combined with variations in individual behaviours and overall stochasticity in transmission often limit the impact of these control

measures. For example, while effective in general, hand-washing may fail due to a few "super-spreader" individuals who do not comply with hygiene recommendations [8]. Additionally, the efficacy of each intervention itself may vary. For vaccination, the efficacy to reduce acquisition is uncertain due to the current absence of such a vaccine on the market [9]. For hand-washing, the efficacy to reduce transmission depends on the type of product used and on the compliance to the intervention [10]. In any case, these variations can affect the choice of which implementation strategy should be considered.

Mathematical modelling can provide key insights on the potential effectiveness of interventions while accounting for these uncertainties and identify which implementation strategies are likely to remain effective across a large range of possible values. For this, however, realistic models are required, able to capture the key transmission dynamics relevant to the setting of interest, and parameterised using appropriate data.

Because the structure of contact networks within healthcare settings influences the spread of HAI pathogens [11], manipulating contact network structures or targeting highly connected individuals may significantly improve the efficacy of control measures [12]. Here, using individual-based modelling of nosocomial pathogen spread, combined with fine-grained longitudinal data on human close proximity interactions (CPIs), we show how detailed knowledge of the structure of human interactions may help design more effective interventions for HAI control. We illustrate this point through an application to control the spread of colonisation by MRSA in a long-term care facility (LTCF).

## Methods

### Data description

Data used here were previously collected during the Individual-Based Investigation of Resistance Dissemination (i-Bird) study [13,14], which took place within a rehabilitation and LTCF from the beginning of July to the end of October 2009. Over this period, each participant (patient or hospital staff) was wearing an RFID sensor that recorded CPIs (at less than 1.5 m) every 30 s. A dynamic network of proximities is therefore available over 117 days with information on individual ID, ward of affectation, age, gender, etc. In addition, dedicated nurses swabbed patients and hospital staff each week to detect MRSA colonisation.

The studied hospital is dedicated to follow-up care and rehabilitation of patients, with no complex procedures being conducted on-site. The hospital was structured into 5 wards: (i) 3 neurology care wards; (ii) 1 nutritional care ward; and (iii) 1 elderly care ward. In addition to patients in neurology, elderly and nutritional care, in-patients also included those in persistent vegetative state (PVS), and those in postoperative and orthopaedic care. Most patients had long hospitalisation durations (median: 7 weeks). In addition to "classic" staff categories such as nurses, physician, rehabilitation staff, patients could interact with other staff members, such as hairdressers. Patients and staff could also interact with individuals belonging to their own group.

Overall, a total of 327 patients and 263 hospital staff had recorded contacts during the investigation period. This study is described in more detail in [13,14].

### Model description

We developed a stochastic Susceptible-Colonised-Susceptible individual-based model that simulates the dynamic transmission of a pathogen within a hospital over a network incorporating data on the detailed structure of CPIs. Individuals could either be patients or hospital staff members. Hospital staff were divided into 6 categories: healthcare assistants, nurses (including nurses, head nurses, and students), rehabilitation staff (occupational therapists, physiotherapists, and other rehabilitation staff), physicians, hospital porters, and other staff (animation,

logistic, administration, and hospital service agents). The model accounts for admissions and discharges from the hospital and interindividual contacts. Once admitted, a patient remains in the hospital until discharged, whereas hospital staff can be present or absent according to their daily schedule. Patient admission, discharges, and staff presence times are all directly taken from the i-Bird study data to reproduce the study conditions in the model.

## Transmission process

Every individual can either be colonised or non-colonised (susceptible) by the pathogen (here, MRSA). At each contact between a susceptible and a colonised individual, the pathogen can be transmitted from the colonised to the susceptible individual with a given probability. This transmission probability is computed as the product of the between-individual contact duration and the pathogen-specific transmission probability, assuming that risks saturate after 1 h. The model accounts for 4 different transmission probabilities depending on the status of the individuals involved: patient-to-patient, patient-to-staff, staff-to-patient, and staff-to-staff (see S1 Text). These probabilities are fixed, but the process of transmission itself is stochastic. To determine if transmission occurs during a contact, a number is randomly drawn from a Uniform distribution between 0 and 1. If the number is lower than the relevant transmission probability, we consider that transmission occurs.

A colonised individual can naturally recover to the susceptible state after a colonisation duration randomly drawn from a lognormal distribution, since this distribution was similar to the observed one (S1 Fig). Such individuals may subsequently be recolonised (no immunity is assumed). We also assume that no active decolonisation measures are implemented.

Individuals are assumed to be screened for colonisation with a probability estimated from the data that depends on weekdays. The total number of swabs for a given day for patients and staff are separately drawn from 2 Normal distributions parameterised using the i-Bird study data, since these distributions were similar to the observed ones (S1 Fig). Patients and staff are then randomly selected to be swabbed among those present in the facility on the given day.

## Model parameterisation

The model was parameterised using the i-Bird data. Simulations ran over 84 days, with an initial 151 patients and 236 hospital staff members present, to reflect the duration and conditions of the data collection. Values for model parameters were directly computed from the observed data on MRSA colonisation among the patients and hospital staff. A summary list of model parameters is provided in Table A in S1 Text. Detailed information on parameter value calculations is provided in S1 Text.

## Building synthetic contacts

We built an algorithm to generate both realistic full and reported stochastic dynamic networks of interindividual interactions in the hospital using parameters estimated from the observed data. Details of parameters computations and CPI generation algorithm are provided in reference [15]. Briefly, the contact network measured in the i-Bird study captures patient–patient, staff–staff, and patient–staff contacts. Contacts are assumed to be reciprocal. In the observed network, not all contacts may have been recorded, due to several possibilities which we cannot distinguish retrospectively (technical issues with sensors failing or low batteries, individuals not wearing the sensors. . .). In parallel to the contact data, presence data was recorded for patients with admission and discharge dates, and for staff based on their work schedules. We developed an algorithm to reconstruct contacts at times when individuals were known to be present in the facility, but had no contacts recorded during that time. We performed this

reconstruction by stratifying individuals by ward and category (either patient or staff type) and used information on contacts recorded for each group (frequency, duration, ward and category of contacted individual) to infer missing contacts for the same group. This reconstruction is stochastic, since for each contact reconstructed, the identity of the contacted individual was randomly chosen among individuals belonging to the target ward and category and present in the facility at that time.

## Assessed control strategies

We evaluated 3 distinct contact-based control strategies: staff reallocation, contact precautions, and vaccination. The way that each intervention impacts the contact network in the transmission model is highlighted in S2b Fig.

Reallocation was simulated as a modification of the contact network, in which patients were allocated to a given staff member of each category for their entire length of stay. In the baseline contact network, patients are not always taken care of by the same members of staff, which increases overall connectivity and therefore pathogen transmission across the entire facility. In the case of the reallocation intervention, we generated contacts using a previously described algorithm [15], choosing in priority the staff member allocated to that patient (or vice versa) when a corresponding contact occurred. For example, if we allocate patient *p1* to nurse *n1*, then nurse *n1* will systematically be chosen in priority whenever the algorithm attempts to create a contact between *p1* and a nurse. Through this, we therefore reduce overall connectivity in the network while maintaining patient care needs (e.g., if in the baseline network patient *p1* interacts with a nurse 3 times each day, then this will still be the case in the reallocation contact network). Importantly, unlike the alternative "patient reallocation/cohorting" intervention which can be found in the literature [16], the reallocation of patients to staff members is made independently of colonisation status; that is, we do not specifically regroup colonised individuals together and assign staff members to only take care of either colonised or uncolonised individuals. We assumed that reallocation did not influence CPI rates. We maintained the global care needs of patients over the entire period, defined by the average number of unique contacts in the data between each patient and different staff categories, by ward. A series of 64 scenarios exploring different combinations of staff categories affected by reallocation were implemented. For each scenario, 30 independent contact networks were stochastically generated in accordance with the new organisation.

Contact precautions were simulated by reducing instantaneous patient-to-hospital staff and hospital staff-to-patient transmissions probabilities 2-, 4-, 6-, 8-, or 10-fold, irrespective of CPI rates. Three specific scenarios were investigated: (i) contact precautions for all members of different staff categories; (ii) contact precautions for 60 randomly selected staff members among nurses or all staff; and (iii) contact precautions for 60 individuals with the highest rates of contacts, called "supercontactors." Two definitions of supercontactors were assessed: (i) based on the number of contacts (henceforth called "frequency-based supercontactors"); and (ii) based on the duration of contact (henceforth called "duration-based supercontactors"). Frequency-based supercontactors were defined as the patients or hospital staff members who had the highest mean number of daily contacts with distinct individuals. Duration-based supercontactors were defined as the patients or hospital staff members who had the highest mean daily cumulative duration spent in contact with other individuals. Several strategies were explored regarding the type (patients and/or staff members) of selected supercontactors on whom to focus reinforced contact precautions.

Vaccination was simulated by reducing acquisition probabilities for vaccinated individuals by 2-, 4-, 6-, 8-, or 10-fold, irrespective of CPI rates. The effect of this hypothetical vaccine

therefore corresponds to an unvaccinated-to-vaccinated transmission probability reduction, regardless of the categories of individuals in contact (staff or patient). For example, a 6-fold reduction would translate into a vaccine efficacy of $1 - 1/6 = 83\%$ to reduce the risk of acquisition. We examined the same scenarios as for contact precautions. We assume that the vaccine has been administered with sufficient time before the simulation, and therefore, do not consider a delay before reaching maximum vaccine efficacy. We also do not account for potentially waning immunity due to the relatively short time period of our simulation. Here, we assume that vaccination only reduces acquisition for vaccinated individuals. Hence, the probability of transmission from a vaccinated individual to a non-vaccinated individual remains the same as the probability of transmission between 2 non-vaccinated individuals.

For all interventions and scenarios, the relative reduction in the cumulative incidence of MRSA colonisation over the entire simulation period was used as an indicator of intervention efficacy. This was calculated by simulating each scenario (including baseline) 500 times and comparing each simulation result with 10 randomly chosen simulations of the baseline scenario, leading to a total of 5,000 comparison points per scenario. We used a Wilcoxon test to check if the median relative reduction in cumulative incidence was significantly different from 0 and derive 95% confidence intervals for the estimates as well as $p$-values.

We used the model to simulate the impact of these control strategies for other pathogens than MRSA. To represent the varying epidemiological characteristics of these pathogens, we either doubled or halved the values for the transmission rate or carriage duration (i.e., infectious period) compared to the values we estimated from the data.

## Results

### A simulated hospital contact network that realistically mimics the observed contact network

We designed a stochastic individual-based model (IBM) to reproduce the realistic dynamic network of within-hospital between-human interactions. The model was calibrated to generate simulated contact networks with the same characteristics as the real network provided by the CPI data (see [15] for details). As shown on Fig 1A, the simulated contact network accurately reproduced real average hourly patterns of patient-to-patient, staff-to-staff and staff-to-patient interactions.

### Observed weekly MRSA incidence is well reproduced by simulations of network-based transmission

The Susceptible-Colonised process implemented into the IBM reproduces the transmission process of a colonising pathogen, here MRSA, in the LTCF (S2a Fig). The model was parameterized to mimic the i-Bird study conditions. Model parameters were not calibrated using a model-fitting approach. Instead, they were calculated directly from the i-Bird study data, which contained information on the presence days of patients and staff, as well as positive and negative MRSA colonisation swabs and, as detailed above, between-human interactions assumed to be opportunities for transmission (see S1 Text). The simulated dynamic contact network described in the previous section was used to mimic between-human interactions and assumed to be the support of MRSA transmission within this LTCF [13,17]. When initializing the model with MRSA carriage of patients and staff as reported by the i-Bird data, the weekly incidence of MRSA colonisation predicted by the model reproduced well the observed trends and weekly incidence over the study period (Fig 1B). For patients, 12/12 of model-generated median incidences were within the margins of error of the i-Bird study estimates, and

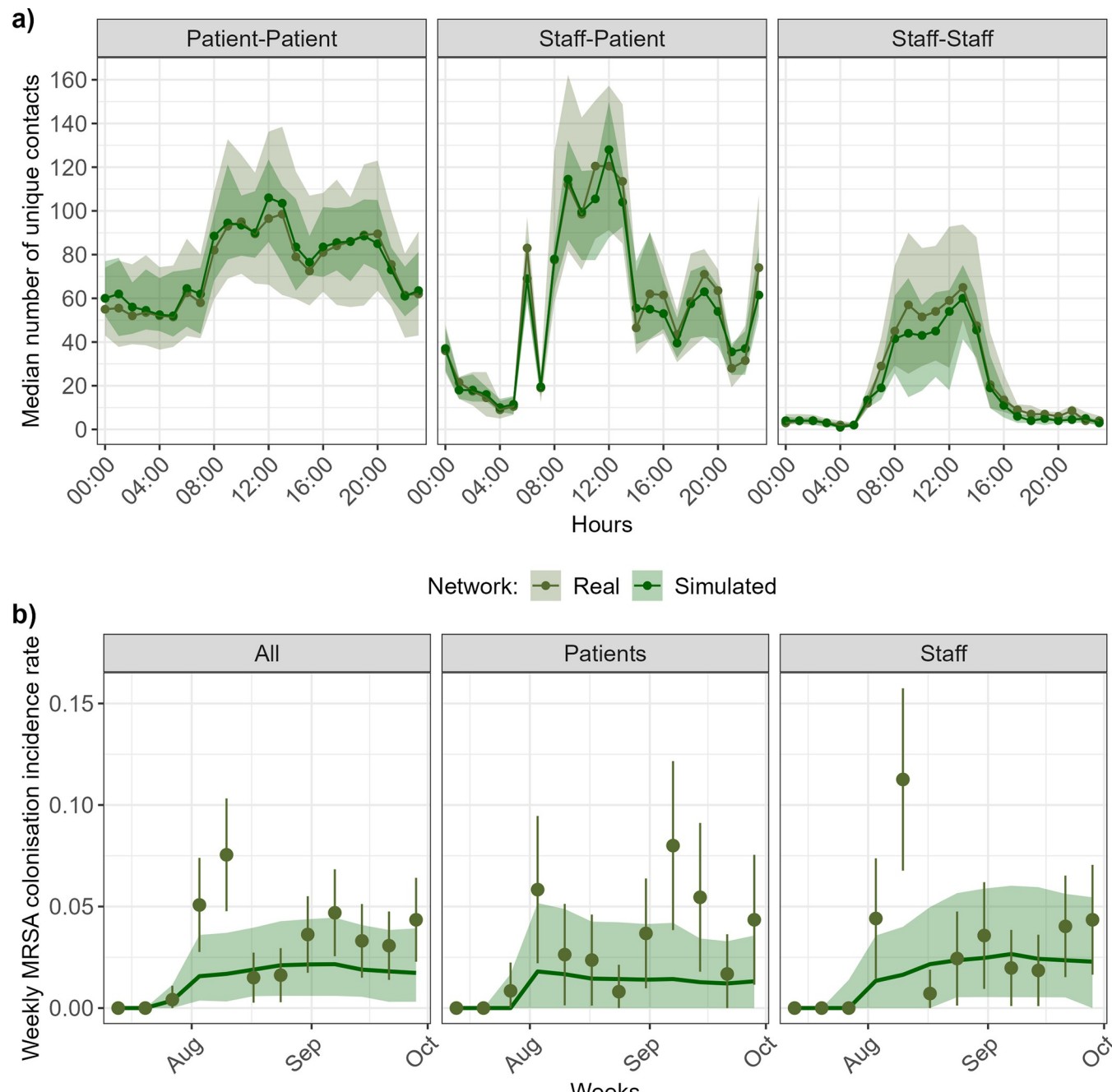

**Fig 1. Real and simulated contacts and MRSA incidence.** (a) Hourly distribution of number of unique contacts. The lines and points show the median estimates, and the shaded areas show the interquartile ranges. The real values come from the i-Bird study, and the simulated values are shown for 50 simulated contact networks. (b) MRSA colonisation weekly incidence over 3 months. Olive points correspond to the observed weekly incidence during the i-Bird study, with lines indicating the margin of error, estimated using the number of individuals swabbed that week. Simulated results are obtained from 15,000 stochastic model simulations (500 simulations of 50 simulated networks). The dark green line shows the median incidence, and the shaded area shows the 95% prediction interval, defined as the interval between the 2.5th and 97.5th percentiles.

8/12 of the observed incidences were within the 95% prediction interval of the model (weeks 4, 9, 10, and 12 being the exceptions). For staff, there are 2/12 outlier weeks where observed incidences were the 95% prediction interval of the model (weeks 4 and 5); all other observed incidences are within the prediction interval. The median cumulative incidence of MRSA

colonisation in the total population of 327 patients and 263 staff over the three-month period predicted by our model was 21 (prediction interval: 11, 31) for patients and 35 (prediction interval: 19, 54) for staff. We use this cumulative incidence as baseline in our analyses on intervention effectiveness in the following sections.

## Hospital staff reallocation, especially in healthcare assistants, reduces MRSA spread

To assess the extent to which the dissemination of MRSA can be restricted through an optimised patient-staff allocation, we first assessed the impact of staff reallocation, defined as the attribution of a reduced number of patients to each staff member during the entire investigation period (S2b Fig).

Simulating the transmission of MRSA over the different networks, we found that reallocation scenarios targeting different hospital staff categories can help reduce cumulative incidence of MRSA colonisation (Fig 2A for scenarios where 1, 2, or all staff categories were reallocated, S3a Fig for all scenarios). Importantly, the benefit of the intervention varied depending on the categories of staff reallocated. When only a single staff category was reallocated, the highest incidence reduction was obtained for healthcare assistant reallocation (median decrease: 40%, 95% confidence interval: 39, 41). All scenarios with 2 categories reallocated involving healthcare assistants prevented between 39% and 56% of colonisations over the entire simulation period. For comparison, reallocating all staff categories prevented 65% of colonisations (CI: 64, 66). Reallocation of either porters or physicians alone barely led to any change in incidence compared to baseline, since these interventions did not substantially change the number of unique staff–patient contacts within the hospital and, therefore, did not substantially affect MRSA spread (S4 Fig). A pseudo-random contact network in which patients were homogenously distributed among all staff members led to more contacts and a higher incidence as compared to the one generated by the baseline network (38% increase, CI: 36, 39), since this increased unique staff–patient contacts within the hospital (S4 Fig).

To see if the variability between scenarios was due to the different number of individuals reallocated in each scenario, we divided the relative incidence reduction for each scenario by the corresponding number of staff reallocated (Fig 2B). This did not substantially change the order of the scenarios with the highest benefit, now calculated as relative incidence reduction per reallocated staff. Scenarios where nurses or healthcare assistants were reallocated remained high in the ranking, despite requiring a large number of staff to be allocated. Reallocation of healthcare assistants only led to the highest overall relative reduction per staff reallocated ($3.9 \times 10^{-3}$%, CI: $3.8 \times 10^{-3}$, $4.0 \times 10^{-3}$), even higher than if all staff are reallocated ($2.4 \times 10^{-3}$%, CI: $2.4 \times 10^{-3}$, $2.5 \times 10^{-3}$). In any case, we still note heterogeneity in the efficacy of different scenarios, indicating that there are other relevant characteristics which differ between staff categories.

## Reinforced contact precautions or vaccination of nurses or healthcare assistants are more effective than staff reallocation

Next, we investigated the impact of reinforced contact precautions taken by hospital staff (e.g., glove wearing or improved hand hygiene compliance) and a hypothetical vaccination intervention (S2b Fig). Contact precautions were simulated as a 2- to 10-fold reduction in both patient-to-hospital staff and hospital staff-to-patient MRSA transmission probabilities during contacts. Vaccination was simulated as a 2- to 10-fold reduction in MRSA acquisition probabilities during contacts between any colonised individual and a non-colonised vaccinated individual.

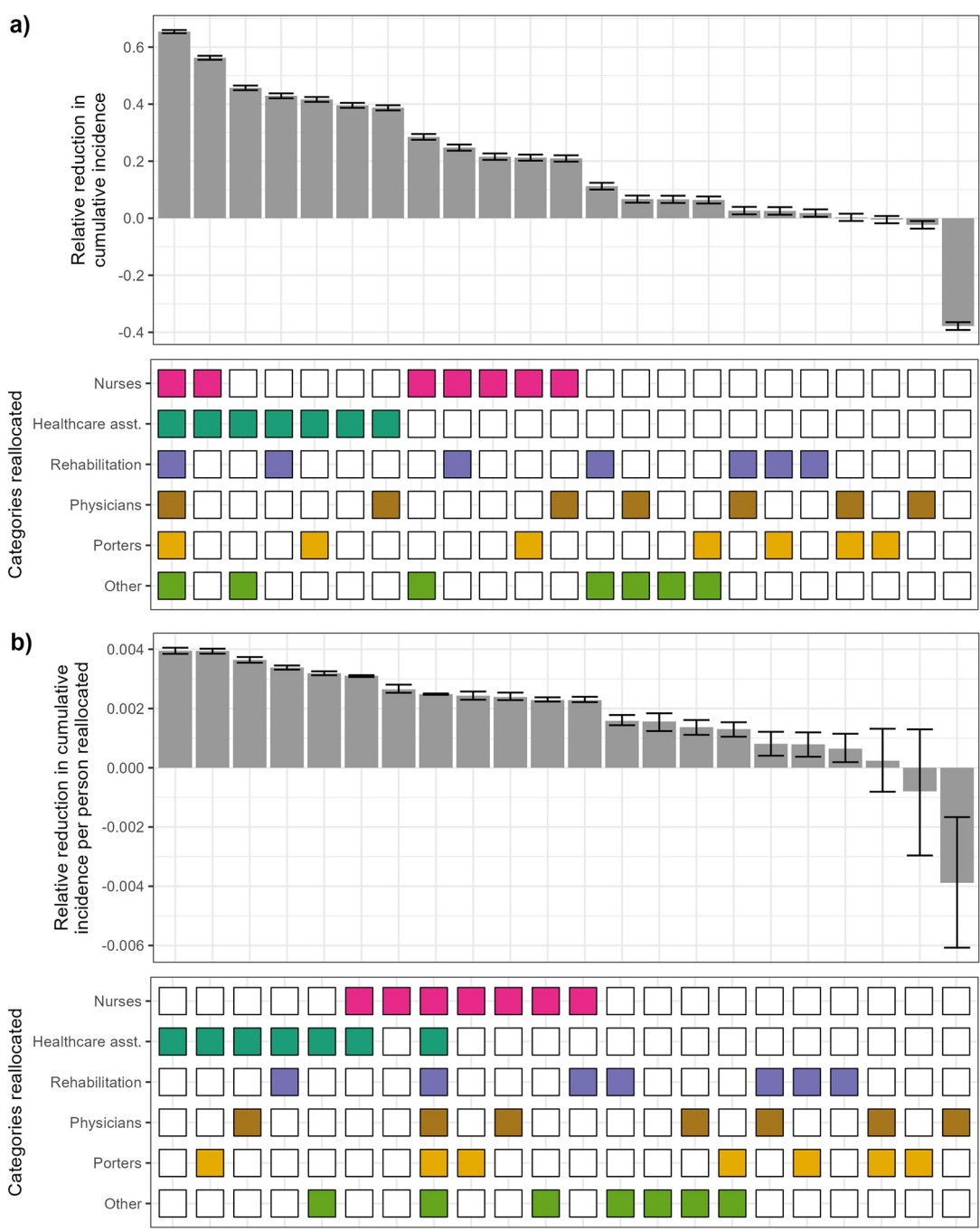

**Fig 2. Relative reduction in cumulative incidence of MRSA colonisation for different hospital staff reallocation scenarios, shown per scenario (a), or per scenario divided by number of staff reallocated in that scenario (b). Top:** Each bar depicts, for a given scenario, the median relative reduction between 500 model simulations with no intervention, and 500 simulations with staff reallocation. Error bars indicate the 95% confidence interval, calculated with Wilcoxon tests to assess if the relative reduction is significantly different from 0. All corresponding $p$-values are <0.001, except where error bars cross 0 ($p$ > 0.05, for reallocation of porters only or porters and physicians). A negative reduction indicates that the intervention led to an increase in cumulative incidence. **Bottom:** In each scenario, staff categories coloured are those reallocated. In the scenario with no coloured squares, the contact network is random. In each plot, the scenarios are ranked from most to least effective.

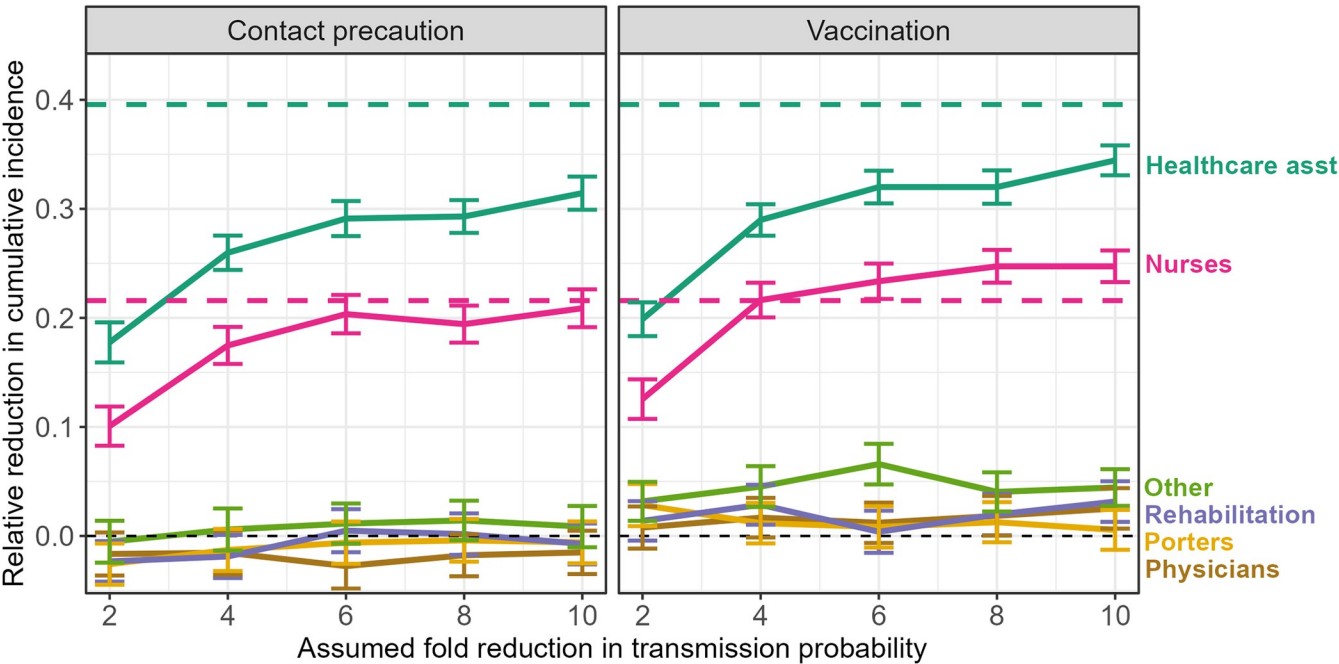

**Fig 3. Effect of contact precautions and hypothetical vaccination targeting different hospital staff categories, compared to staff reallocation.** The coloured dashed lines show the median reduction when reallocating healthcare assistants only (turquoise) or nurses only (pink). All other estimates are shown as median calculated for 500 intervention simulations. Error bars indicate the 95% confidence interval, calculated with Wilcoxon tests to assess if the relative reduction is significantly different from 0. All corresponding $p$-values are <0.001, except where error bars cross 0 ($p > 0.05$).

Contact precautions targeting healthcare assistants led to a large reduction in MRSA colonisations, ranging from 18% to 31% as the assumed level of reduction in transmission probabilities increased from 2- to 10-fold (Fig 3). This was closely followed by contact precautions targeting nurses (10% to 21% reduction). Contact precautions for nurses appear to be as effective as reallocation of this staff category alone, as even an assumed 4-fold reduction in transmission probabilities was sufficient to achieve a decrease in incidence equivalent to reallocation (Fig 3). Vaccination of healthcare assistants or nurses had higher impact than contact precautions (Fig 3).

By opposition, contact precautions or vaccination focused exclusively on either hospital porters, physicians, rehabilitation, or other staff appeared ineffective, with percent reductions below 5% irrespective of the assumed transmission probability reduction (Fig 3).

### Heterogeneous distribution of "supercontactors" among patients and staff

To understand why intervention effectiveness to reduce the spread of MRSA varied depending on the staff category targeted, we examined the extent to which different individuals were connected in the contact network. We identified individuals substantially more connected than others, and henceforth refer to them as "supercontactors." We distinguish between 2 types of supercontactors: (i) individuals with the highest number of daily distinct contacts (henceforth called "frequency-based supercontactors"); and (ii) individuals with the highest overall daily contact duration (henceforth called "duration-based supercontactors").

We identified the top 60 duration- and frequency-based supercontactors for both patients and staff (i.e., top 20% of individuals). If all individuals had the same probability of being supercontactors, we expect that the distribution of patients/staff categories among

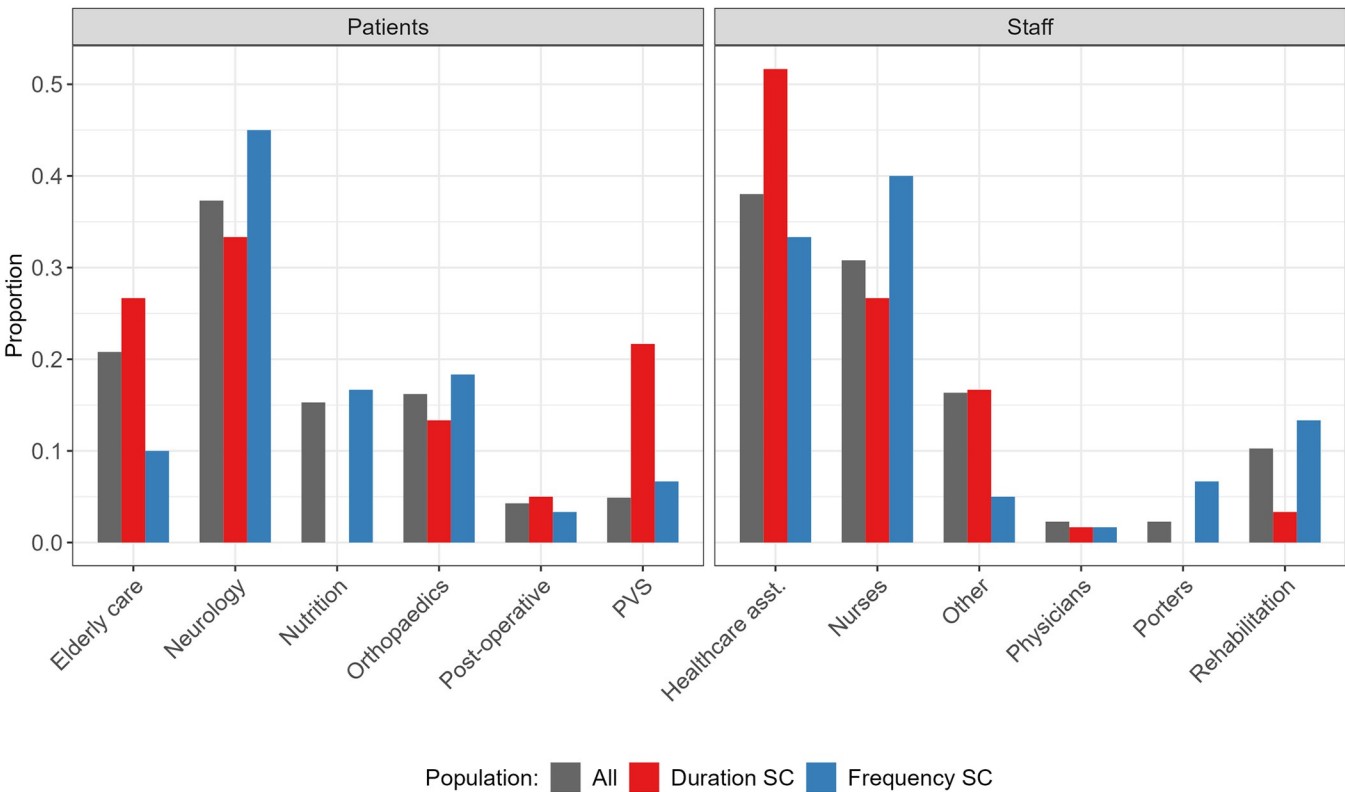

**Fig 4. The distribution of supercontactors (SC) among hospital patients and staff is not homogeneous.** The grey bars show the distribution of categories among all patients (left) or staff (right), the red bars show the distribution of duration-based supercontactors, the blue bars show the distribution of frequency-based supercontactors. If supercontactors were homogeneously distributed among categories, all the coloured bars would be aligned with the grey bars. Here, only the distribution of the top 60 frequency-based and duration-based supercontactors for patients and staff is shown. PVS, persistent vegetative state.

supercontactors (Fig 4, red and blue) would be aligned with the distribution of those same categories among all patients/staff (grey).

Among patients, those in neurology care are the first category of supercontactors (Fig 4, left; 34% of duration-based, 45% of frequency-based). The observed distribution of patient categories among duration-based supercontactors (red) differed significantly from the distribution of those categories among all patients (grey; log likelihood ratio test: $p$-value <0.001). We observed a greater proportion of patients in PVS and elderly care among duration-based supercontactors than among all patients (Fig 4, left). The difference was not statistically significant for frequency-based supercontactors (log likelihood ratio test: $p$-value >0.2).

Among staff, the majority of supercontactors were either healthcare assistants (Fig 4, right; 52% of duration-based, 33% of frequency-based) or nurses (Fig 4, right; 26% of duration-based, 40% of frequency-based). The observed distribution of staff categories among supercontactors differed significantly from the distribution of those categories among all staff (log likelihood ratio test: duration-based $p$-value <0.01, frequency-based $p$-value <0.05). Compared to the distribution among all staff, we observed a greater proportion of healthcare assistants among duration-based supercontactors and a greater proportion of nurses, porters, and rehabilitation staff among frequency-based supercontactors (Fig 4).

There was almost no overlap between the identities of the frequency- and duration-based supercontactors. Only 3 patients in PVS, 2 patients in neurological care, 1 nurse, and 1 rehabilitation staff were in both categories.

## Targeting supercontactors is most effective to reduce MRSA spread

We used supercontactors as target for interventions in the hospital. We compared the effect of reinforced contact precautions or hypothetical vaccination, targeting different combinations of 60 supercontactors (i.e., contact-based or duration-based supercontactors among both patients or hospital staff), 60 staff randomly chosen, or 60 patients randomly chosen. Here, we only show the reductions for an assumed 6-fold reduction in transmission probabilities, with other fold reductions shown in S5 Fig.

Targeting supercontactors within either staff or patients with an intervention was at least as effective to reduce incidence than randomly targeting individuals in the same group with the same intervention (grey, Fig 5). When targeting staff, implementing an intervention on a mix of frequency- and duration-based supercontactors (purple) was most effective to reduce the cumulative incidence of MRSA colonisations across the facility. When targeting patients, focusing on duration-based supercontactors (red) gave better results. Regardless of the type of supercontactors targeted, vaccination was equally or more effective than contact precautions (Fig 5). Overall, vaccinating a mix of duration- and frequency-based staff supercontactors appeared to be the most effective, with up to 23% (CI: 22, 24%) of colonisations prevented, closely followed by vaccinating a mix of staff and patient duration-based supercontactors.

These results are partly explained by the relative time spent by staff and patients in contact with either individuals of the same group (e.g., staff with staff) or of a different group (e.g., staff with patients) (S6a Fig), as well as the relative values of the transmission probabilities for each type of contact (S6b Fig and Table A in S1 Text). This is because, in our model, contact precautions can only reduce between-group transmission (staff-to-patient and patient-to-staff), while

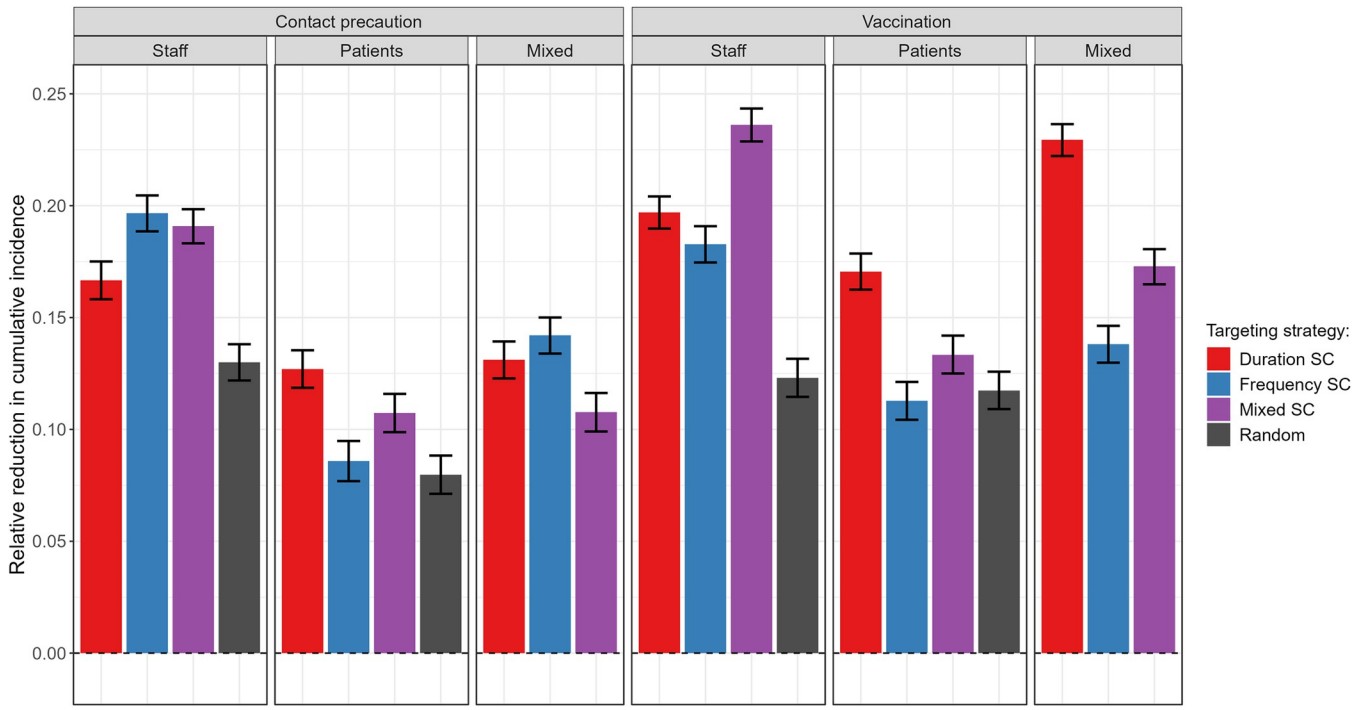

**Fig 5. Comparison of contact precautions or hypothetical vaccination for 60 staff, patients, or a mix of staff and patients, targeting either duration-based supercontactors (SC), frequency-based SC, a mix of duration and frequency-based SC, or random individuals.** We assume the interventions lead to a 6-fold reduction in transmission probabilities. For each strategy, the bar indicates the median relative reduction in cumulative incidence obtained for 500 simulations. Error bars indicate the 95% confidence interval, calculated with Wilcoxon tests to assess if the relative reduction is significantly different from 0. All relative reductions were significantly different from 0 ($p < 0.001$).

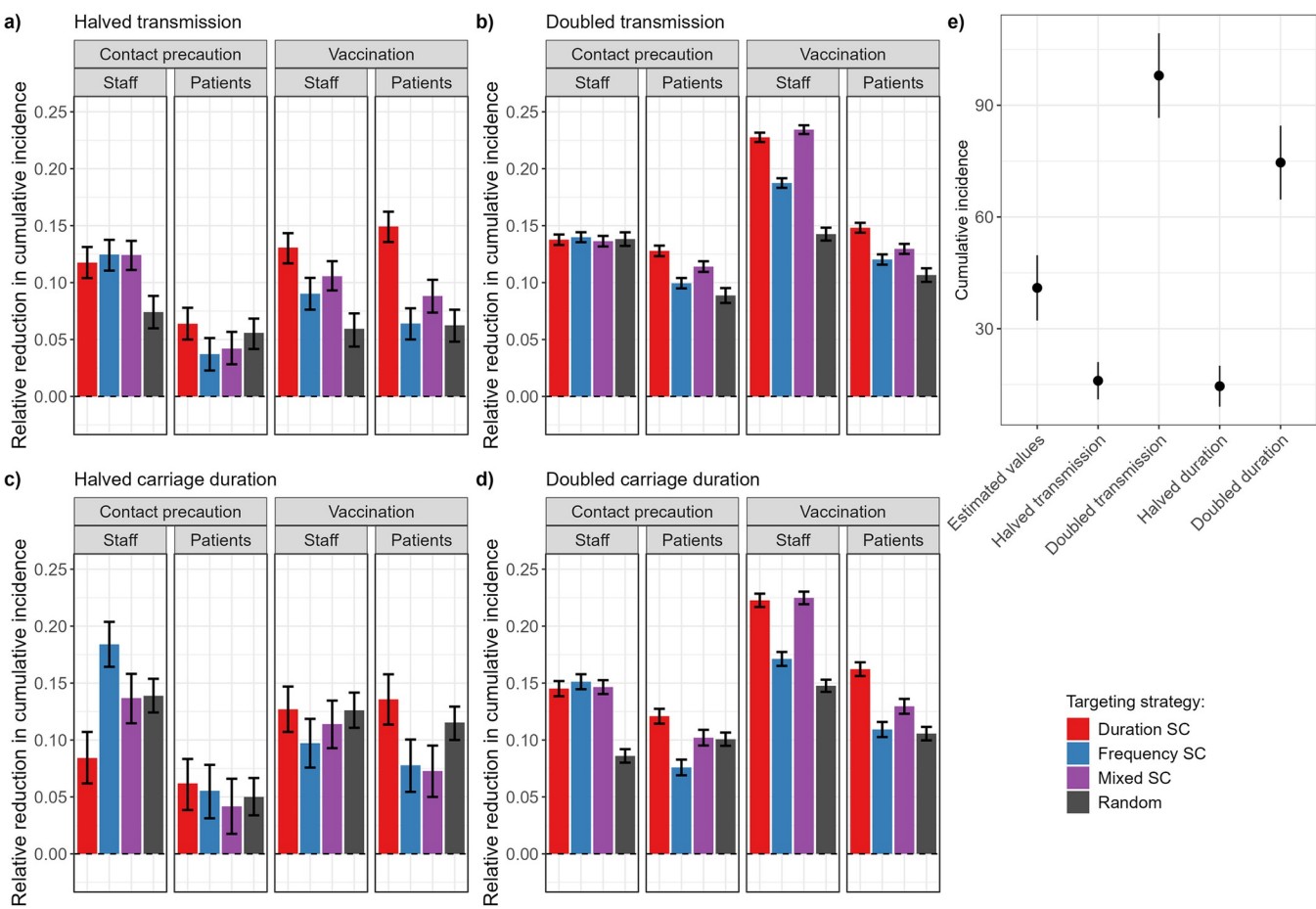

**Fig 6. Comparison of contact precautions or vaccination for 60 staff or patients, targeting either duration-based supercontactors (SC), frequency-based SC, a mix of duration and frequency-based SC, or random individuals, and varying either the baseline transmission rate or carriage duration.** (a) Halved transmission rate; (b) doubled transmission rate; (c) halved carriage duration; (d) doubled carriage duration. We assume the interventions lead to a 6-fold reduction in transmission probabilities. For each strategy, the bar indicates the median relative reduction in cumulative incidence obtained for 500 simulations. Error bars indicate the 95% confidence interval, calculated with Wilcoxon tests to assess if the relative reduction is significantly different from 0. All relative reductions were significantly different from 0 ($p < 0.001$). (e) Absolute cumulative incidence without intervention using estimated parameter values, higher/lower transmission, or higher/lower carriage duration. Points indicate the mean, and lines mean +/− standard deviation, obtained for 500 simulations.

vaccination reduces acquisition and hence can reduce within-group transmission (staff-to-staff and patient-to-patient) (S2 Fig). For example, for patients, vaccination may be more effective than contact precautions since it reduces the largest per-contact transmission probability (staff-to-patient) and reduces the probability for the most dominant type of contact (patient-to-patient) (S6 Fig).

These conclusions are maintained when assessing different contact precautions or vaccination efficacies, i.e., assuming 2-, 4-, 8-, or 10-fold reductions in transmission or acquisition probabilities, respectively (S5 Fig), or when targeting 20 or 100 individuals instead of 60 (S7 Fig).

## Targeting supercontactors is also an effective strategy for other nosocomial pathogens

Although the epidemiological parameters we used in the previous sections were directly estimated using data on MRSA, our model can be applied to any nosocomial pathogen for which

CPIs are the main vector of transmission, as opposed to other environmental routes of transmission such fomites. Naturally, the epidemiology of such pathogens would likely vary compared to MRSA, with different transmission rates and carriage/infectiousness durations compared to the values we estimated. To investigate the applicability of our results to other pathogens, we repeated our analysis above, doubling or halving either the transmission rates or the carriage/infectiousness durations. Our qualitative results on the value of targeting supercontactors to improve intervention effectiveness remained valid (Fig 6A–6D), even with different baseline incidences due to the parameter changes (Fig 6E). Interestingly, we see that in a few scenarios targeting patients or staff randomly could be slightly more effective than targeting frequency-based patient supercontactors (Fig 6A–6D). This is due to the high effectiveness of targeting duration-based supercontactors in such instances, combined with the nonoverlapping identities of duration- and frequency-based supercontactors. Inevitably, by exclusively targeting frequency-based supercontactors we exclude duration-based supercontactors, while random targeting may still incidentally include these individuals. The reverse is observed for staff contact precautions at halved carriage duration, where random targeting is more effective than targeting duration-based supercontactors (Fig 6C).

## Discussion

In this study, we present how the dynamic interindividual contact network of a healthcare institution can be analysed to implement efficient interventions aimed at reducing pathogen transmission. We first applied an IBM to a French LTCF and confirmed that it reproduced well both the recorded network and MRSA dynamics. We then evaluated and compared several network-based control strategies, demonstrating that while hospital staff reallocation can help reduce MRSA transmission overall, staff contact precautions and hypothetical vaccination could be as or more effective than reallocation. Interestingly, the efficacy varied depending on which staff category was targeted by the intervention. We identified "supercontactors" in the contact network with more or longer contacts and found that these were heterogeneously distributed among staff and patient categories. The effectiveness of contact precautions and vaccination was further increased by targeting these supercontactors in the LTCF, compared to randomly targeting individuals. Our conclusions remained valid when varying epidemiological parameters, suggesting that targeting supercontactors is also an effective strategy for other nosocomial pathogens transmitted via CPIs. Importantly, our work shows that while contact precautions and vaccination both act by reducing transmission probabilities, their impact can differ since the specific probabilities and contact types they affect may not be the same. Therefore, understanding the relative importance of these elements and their contribution to the overall contact network dynamics is crucial to implement efficient interventions against pathogen spread.

Here, we demonstrated that staff reallocation is an effective strategy to reduce transmission risk. Moreover, reallocation strategies involving healthcare assistants were the most effective. Our simulation results are consistent with previous work on this topic, showing the best staff reallocation strategies were those significantly lowering the degree of the hospital worker-to-patient subgraph [12,16,18–23]. In particular, our results are in agreement with those of Mietchen and colleagues, who used a compartmental model to show that assigning each nurse to a specific group of patients (similar to our staff reallocation intervention) reduced MRSA acquisition by 50% compared to when patients and staff interacted randomly [24]. In a previous study, we examined the potential of different hospital staff categories to spread nosocomial pathogens and to play a role of super-spreader, showing the importance of adherence to contact precautions in "peripatetic" hospital staff. These later were defined as hospital staff

members with relatively short contacts, but with many patients, a definition similar to the "frequency-based supercontactors" here [8].

Since transmission was modelled through the contact network, supercontactors can mechanistically play the role of super-spreaders, but also be themselves more at risk of acquiring the bacteria during a contact with a colonised individual. These factors explain why targeting supercontactors for interventions led to a substantial reduction in colonisation incidence. The most appropriate supercontactor type to target (duration-based or frequency-based) surprisingly differed between patients and hospital staff: while targeting a mix of duration- and frequency-based supercontactors was more relevant for hospital staff, duration-based supercontactors were selected for patients. We also predicted that the most effective intervention to reduce the overall incidence of colonisation was to vaccinate a mix frequency- and duration-based supercontactors among staff with a hypothetical vaccine, which here we assume protects against acquisition. These conclusions may be related to the relative importance of the within-group transmission probabilities (patient-to-patient and staff-to-staff), as well as the high proportion of total contact time supercontactors spend in contact with individuals of the same group they belong to which we observed in the i-Bird data (S6 Fig). Since vaccination limits within-group transmission, while contact precautions only affect between-group transmission, this partly explains why vaccination was more effective. These results may be specific to the type of hospital investigated here. In LTCF, the frequency and duration of patient–patient interactions are much higher than in acute care facilities. Our results highlight the necessity of involving patients in intervention implementation in LTCF.

It is important to note that the hospital followed up during the i-Bird study included neurology care wards hosting patients in PVS. These patients in PVS accounted for one fifth of the individuals classified as duration-based supercontactors (Fig 3). While they may be considered similar to sedated and ventilated patients in intensive care units, the presence of this type of patients with particularly long contacts and specific behaviours is not universal across all types of LTCF. To improve the generalisability of our results to other LTCF, we performed an additional analysis in which patients in PVS were excluded when identifying supercontactors: this hypothesis did not affect our conclusions (S8 Fig).

The results presented here should be interpreted in the light of the following limits. Firstly, we only considered here that MRSA transmission occurred through interindividual contacts among participants, with a risk of transmission saturating after 1 h. This assumption was based on previous analysis of the same data, suggesting that the proximity network was the main transmission route for MRSA acquisition in this setting [13]. In this study, while participation was high (95% of staff and patients agreed to wear the sensors), it was also estimated that 25% of MRSA acquisitions were not explained by the contact network, and may instead be mediated by other acquisition mechanisms not included in our model, such as environmental contamination, or bacterial evolution within the host leading to the emergence of resistance. Importations of new colonisations, through for example hospital visitors or patient's permissions outside the hospital were also not included in the model, while they could also have been sources of MRSA acquisition during the i-Bird study. This may explain why model simulations slightly underestimated the incidence point on the fifth week, as illustrated in Fig 1B.

Secondly, we did not account for the infection status of patients in the model. Over the study, several infections occurred in participating patients (eschar, cutaneous infection, gastrostomy, colostomy, tracheotomy, ulcer, etc.). When an infection occurs, bacterial load is usually much higher, which could potentially increase the risk of bacterial dissemination in the environment or transmission to contacts. Infections could also impact the dynamic of contacts and of nurse scheduling, as infected patients are bound to have a higher care load, thus requiring

more contacts. Interestingly, this higher care load could reclassify infected patients as super-contactors and, as we have shown here, identify them as key targets for interventions to reduce spread. For these reasons, future work taking into consideration infected patients may further improve our ability to implement effective interventions.

Thirdly, the epidemiological parameters of the model, which included transmission probability and carriage duration, were directly estimated for MRSA from the admission, schedule, swab, and contact data [13,14]. While this allows us to be confident that our model reproduces well the characteristics of the LTCF in which the i-Bird study was conducted, these parameters can vary depending on the estimation period (e.g., holidays versus term-time), setting (e.g., long term versus acute care), population (e.g., older versus younger), and circulating bacterial or viral pathogen in the hospital. For example, the probability of MRSA transmission that we estimated is slightly lower than in other studies (e.g., 0.000030 per 30 s of contact on average for hospital staff-to-hospital staff and 0.000722 for hospital staff-to-patient in our study with the real RFID network, compared to a probability between 0.0005 and 0.0050 per 30 s of contact in the study by Hornbeck and colleagues [25]). The durations of MRSA colonisation that we estimated from the data (28 days for patients, 18 days for hospital staff) are also either shorter or longer than previously reported estimates, but these values can be clone or setting specific [26,27]. Among other pathogens transmitted by CPI, *Klebsiella pneumoniae* has characteristics within the range we explored in our analysis (transmission probability of 0.0005 per 30 s of contact, carriage duration of 3 weeks) [28]. SARS-CoV-2 is another example with a similar transmission probability, although the infectious period (equivalent to the carriage duration) is lower (9 days) [29]. As we have shown, our conclusions on the value of interventions strategies targeting supercontactors were not impacted by changes in parameters to reflect the epidemiology of these other pathogens instead of MRSA. However, these conclusions would likely not be directly applicable to pathogens for which CPIs are not the main route of transmission. For ESBL-producing *E. coli*, previous work has shown that the contact network measured in the i-Bird study could not accurately capture transmission, by opposition to methicillin-susceptible and -resistant *S. aureus* and ESBL-producing *K. pneumoniae* [13,28].

More generally, our use of the i-Bird study data here is both a strength and limitation of our approach. By aiming to reproduce the specific incidence in this setting using data to directly calculate relevant parameters, we were able to rely on minimal parameter assumptions. We are therefore confident that our model is representative of the real situation in the LTCF, adding more credibility to our conclusions on the potential impact of interventions in this setting. However, we also recognise that the characteristics of one LTCF may not be representative of all LTCFs, and that these characteristics may even change within a single LTCF over time. As mentioned in previous paragraphs, we tried to account for this by showing that our conclusions remain valid even by changing several key characteristics, such as the presence of patients in PVS or the estimated epidemiological parameters. That said, there are some elements we have not explored such as different care organisation which may be found in other countries and settings. We expect that our overall conclusions pertaining to "supercontactors" would still hold under different conditions, since contacts will always follow a similar heterogeneous distribution, with some individuals being in contact with more individuals or spending more cumulative time in contact than others. To the best of our knowledge however, no study has yet compared such distributions across settings.

Lastly, in this analysis we explored a wide range of possible intervention efficacies rather than focus on single "likely" values. In the case of contact precautions, efficacy may indeed vary depending on the setting characteristics and individual compliance to the intervention [10]. For vaccination, here we focused on a hypothetical vaccine since no vaccine is currently publicly available against *S. aureus*. However, due to the high burden of this pathogen and the

number of vaccines in clinical development [9], we believe it is still worthwhile to investigate the potential impact of this intervention and clarify its value compared to existing ones such as contact precautions, over a range of possible efficacies. For further generalisability, we examined which intervention strategy targeting either staff or patients is the most effective when the efficacies of contact precautions and vaccination are allowed to vary independently of each other (S9 Fig). This does not affect our conclusions and vaccination remains the most effective intervention in most cases (targeting duration-based supercontactors for patients, and a mix of duration- and frequency-based supercontactors for staff), except if the efficacy of vaccination is assumed to be extremely low (i.e., 2-fold reduction in transmission probabilities or less) while the efficacy of contact precautions is greater (at least 4-fold reduction) (S9 Fig).

Despite their limitations, mathematical models are powerful tools to inform the efficacy of control strategies in hospital settings [30], when they are based on a good understanding of pathogen transmission routes and heterogeneity in human interactions [31,32]. Over the last decades, different approaches have been used to acquire knowledge on interindividual contacts, such as observational studies, diaries, interviews, and more recently wearable sensors [33–41]. While several IBMs of pathogen spread within hospitals [31,42–50] have been developed to assess measures such as hygiene compliance [25] or antiviral prophylaxis impact on influenza [51], few models have actually attempted to directly integrate such rich empiric data. To our knowledge, only 2 published IBMs simulated transmission along an RFID-based contact network [13,25], one of which studied MRSA spread [25]. In that work, Hornbeck and colleagues showed that the number of colonised patients increased when the most connected nurses did not comply with infection control recommendations, which is consistent with our results.

We must consider the feasibility, cost, and social acceptability when deciding which control strategies should be implemented. For example, we suggest that the best strategy would be to implement contact precautions or vaccination focusing on supercontactors, but identifying and targeting supercontactors, in particular among patients, may not be as socially acceptable as broadly targeting hospital staff categories. The benefit of patient vaccination, which we identified as the best strategy in the LTCF, may also be reduced in acute care settings, due to shorter patient lengths of stay and to the likely delay required for immunity to develop following vaccination. In addition, here we chose to simulate the impact of the hypothetical vaccine as a reduction in the probability of colonisation since, although no vaccine is currently publicly available, this type of vaccine is being studied [52–54]. However, we acknowledge that future vaccines may only reduce the risk of infection rather than colonisation [9]. Nonetheless, there are precedents for bacterial vaccines which reduce both infection and colonisation rates, with the most well-known examples being *S. pneumoniae* vaccines [55]. Our analysis could be extended to allow the vaccine to impact different transmission probabilities, such as reducing the risk of transmission from vaccinated individuals compared to non-vaccinated ones. However, since we have shown here that, when targeting the same individuals, vaccination may have a greater impact than contact precautions to reduce overall MRSA colonisation incidence despite only reducing acquisition probability, further reductions in other transmission probabilities would only further reinforce this conclusion.

In any case, achieving a 10-fold reduction in transmission probabilities with either contact precautions or vaccination might not actually be feasible, depending on the baseline level of pathogen transmission, which is why we explored a range of reductions as previous studies have done [56]. On the other hand, reallocation requires greater logistical efforts and may not always be feasible depending on the economic context of the healthcare institution and the care load. Finally, the most effective reallocation strategies may not be the most "cost-effective." For example, when considering the relative reduction in incidence per staff reallocated,

targeting only rehabilitation staff ranked higher than targeting all staff (Fig 2B). These factors will also be affected by the choice of outcome evaluated. Here, we examined the reduction in all acquisitions generated by the interventions, but arguably reducing patient acquisition may weigh more from a public health perspective than reducing staff acquisition, due to the subsequent risk of nosocomial infections.

In conclusion, this work sheds light on the importance of targeting control and prevention measures in an LTCF towards specific hospital staff categories, but also of involving patients in such efforts as they may too play an important role in the transmission network. Patients need to be actors of their own prevention especially when their length of stay is long. More importantly, we underline how monitoring contacts can be helpful to design highly effective control strategies aimed at "supercontactor" individuals.

## Supporting information

**S1 Fig. Distributions of (a) colonisation durations and (b) swabs per day.** The observed distributions in pink are the smoothed densities of observations in the i-Bird study data, while the simulated distributions in blue are the smoothed densities generated by Lognormal (for colonisation durations, a)) or Normal (for number of swabs, b)) distributions informed by the mean and variance of the data.
(TIF)

**S2 Fig. Model outline. (a)** Baseline model description and disease natural history. **(b)** Mode of action of the 3 different interventions examined in the model.
(TIF)

**S3 Fig. Relative reduction in cumulative incidence of MRSA colonisation for different hospital staff reallocation scenarios, shown per scenario (a), or per scenario divided by number of staff reallocated in that scenario (b). Top:** Each bar depicts, for a given scenario, the median relative reduction between 500 model simulations with no intervention, and 500 simulations with staff reallocation, along with the 95% confidence interval. A negative reduction indicates that the intervention led to an increase in cumulative incidence. **Bottom**: In each scenario, staff categories coloured are those reallocated. In scenario 64, the contact network is random. In each plot, the scenarios are ranked from most to least effective.
(TIF)

**S4 Fig. Number of unique patients in contact with each staff member in the baseline network, compared to reallocation scenarios involving either one staff category at a time, all staff, or a random allocation.**
(TIF)

**S5 Fig. Comparison of contact precautions or vaccination targeting 60 individuals, either selected randomly among staff or patients, or different groups of supercontactors, assuming a fold-reduction in transmission probability of (a) 2 or (b) 10.** For each strategy, the bar indicates the median relative reduction in cumulative incidence, with 95% confidence interval, obtained for 500 simulations.
(TIF)

**S6 Fig. Intervention effectiveness depends on between- and within-group contact rates and transmission probabilities. (a)** Most of the cumulative contact time of patient supercontactors is with other patients, while staff supercontactors spend approximately the same amount of time in contact with either patients or staff. **(b)** Contact precautions and vaccination do not reduce the same transmission probabilities. For example, for patients, vaccination may be

more effective than contact precautions since it reduces the largest per-contact transmission probability (staff-to-patient) and reduces the probability for the most dominant type of contact (patient-to-patient).
(TIF)

**S7 Fig. Comparison of contact precautions or vaccination targeting (a) 20 or (b) 100 individuals, either selected randomly among staff or patients, or different groups of supercontactors, assuming a 6-fold reduction in transmission probabilities.** For each strategy, the bar indicates the median relative reduction in cumulative incidence, with 95% confidence interval, obtained for 500 simulations.
(TIF)

**S8 Fig. Comparison of contact precautions or vaccination targeting 60 individuals, either selected randomly among staff or patients, or different groups of supercontactors excluding persistent vegetative state patients, assuming a 6-fold reduction in transmission probabilities.** For each strategy, the bar indicates the median relative reduction in cumulative incidence, with 95% confidence interval, obtained for 500 simulations.
(TIF)

**S9 Fig. Most effective intervention strategy depending on type of individual targeted (patients or staff) and assumed efficacy of each intervention to reduce the relevant transmission probabilities.** For each combination of assumed vaccine efficacy (x-axis) and contact precautions efficacy (y-axis), the colour of the square indicates which type of supercontactor should be targeted to achieve the highest reduction in MRSA colonisation, and the letter indicates which intervention should be used for this purpose. For example, if we are targeting staff (right panel) with either vaccination leading to a 4-fold reduction in transmission probabilities ($x = 4$) or contact precautions leading to a 10-fold reduction ($y = 10$), the best intervention is to vaccinate (V) a mix of duration- and frequency-based supercontactors (purple).
(TIF)

**S1 Text. Detailed CTCmodeler description.**
(PDF)

## Acknowledgments

We would like to thank Christian Brun-Buisson for helpful discussions on the clinical relevance of our results. We acknowledge the help of the HPC Core Facility of the Institut Pasteur for this work.

This communication reflects the author's view and that neither IMI nor the European Union, EFPIA, or any Associated Partners are responsible for any use that may be made of the information contained therein.

## Author Contributions

**Conceptualization:** Quentin J. Leclerc, Audrey Duval, Didier Guillemot, Lulla Opatowski, Laura Temime.

**Data curation:** Quentin J. Leclerc, Audrey Duval.

**Formal analysis:** Quentin J. Leclerc, Audrey Duval.

**Funding acquisition:** Didier Guillemot.

**Investigation:** Quentin J. Leclerc, Audrey Duval, Lulla Opatowski, Laura Temime.

**Methodology:** Quentin J. Leclerc, Audrey Duval, Lulla Opatowski, Laura Temime.

**Software:** Quentin J. Leclerc, Audrey Duval.

**Supervision:** Didier Guillemot, Lulla Opatowski, Laura Temime.

**Validation:** Didier Guillemot, Lulla Opatowski, Laura Temime.

**Visualization:** Quentin J. Leclerc, Audrey Duval.

**Writing – original draft:** Quentin J. Leclerc, Audrey Duval.

**Writing – review & editing:** Quentin J. Leclerc, Audrey Duval, Didier Guillemot, Lulla Opatowski, Laura Temime.

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
