## [Editor Report · Decision Letter 0]

12 Jan 2024

Dear Dr Leclerc, 

Thank you for submitting your manuscript entitled "Using contact network dynamics to implement efficient interventions against pathogen spread in hospital settings" for consideration by PLOS Medicine.

Your manuscript has now been evaluated by the PLOS Medicine editorial staff and I am writing to let you know that we would like to send your submission out for external peer review.

Please re-submit your manuscript within two working days, i.e. by Jan 16 2024 11:59PM.

Feel free to email me at pdodd@plos.org or the team at plosmedicine@plos.org if you have any queries relating to your submission.

Kind regards,

Philippa Dodd, MBBS MRCP PhD

PLOS Medicine

---

## [Decision Letter · Decision Letter 1]

6 Mar 2024

Dear Dr. Leclerc,

Many thanks for submitting your manuscript "Using contact network dynamics to implement efficient interventions against pathogen spread in hospital settings (PMEDICINE-D-24-00101R1)” to PLOS Medicine. The paper has been reviewed by three subject experts and a statistician; their comments are included below and can also be accessed here: [LINK]

As you will see, the reviewers were positive about the paper but, they raised a number of questions about specific study details and the methodological approach. After discussing the paper with the editorial team and an academic editor with relevant expertise, I’m pleased to invite you to revise the paper in response to the reviewers’ comments. We plan to send the revised paper to some of all of the original reviewers*, and of course we cannot provide any guarantees at this stage regarding publication.

When you upload your revision, please include a point-by-point response that addresses all of the reviewer and editorial points, indicating the changes made in the manuscript and either an excerpt of the revised text or the location (eg: page and line number) where each change can be found. Please submit a clean version of the paper as the main article file and a version with changes marked should as a marked-up manuscript. Please also check the guidelines for revised papers at http://journals.plos.org/plosmedicine/s/revising-your-manuscript for any that apply to your paper.

We ask that you submit your revision by March 27th 2024. However, if this deadline is not feasible, please contact me by email, and we can discuss a suitable alternative.

Please don’t hesitate to contact me directly with any questions (pdodd@plos.org). If you reply directly to this message, please be sure to ‘Reply All’ so your message comes directly to my inbox.

Kind regards,

Pippa

Philippa Dodd, MBBS MRCP PhD

PLOS Medicine

plosmedicine.org

pdodd@plos.org

*Please note: If your article is accepted, you may have the opportunity to make the peer review history publicly available. The record will include editor decision letters (with reviews) and your responses to reviewer comments. If eligible, we will contact you to opt in or out.

Editorial comments:

1) We think that your paper could offer a valuable contribution but as written it is not without limitations which we require you address prior to publication. Importantly when revising your manuscript please take care to ensure that it is presented in a manner that is accessible to the general reader as many of our readers will not be experts in infectious disease modelling. In particular, we found your figures difficult rather difficult to digest. 

2) We agree with the reviewers that better contextualisation of your findings in terms of real-world applicability would greatly improve the paper and that clearly defining the novel aspects of your study is required. In that context it would also be helpful to clearly define the long-term care facilities that you refer to – this can mean different things to different people in different parts of the world – much of the existing literature focusses on various hospital settings.

3) We agree that the data pertaining to the hypothetical vaccine can be included in the main paper (PLOS Medicine has published many studies which model hypothetical vaccines) but as per the reviewers and the academic editor, please ensure that additional nuance is included.

4) Of all authors who submit modelling studies we ask that the following points are included in the main manuscript. Please review the list below, derived from Geoffrey P Garnett, Simon Cousens, Timothy B Hallett, Richard Steketee, Neff Walker. Mathematical models in the evaluation of health programmes. (2011) Lancet DOI:10.1016/S0140-6736(10)61505-X and ensure that each item is included in the relvant sub-sections of your manuscript:

* Please provide a diagram that shows the model structure, including how the disease natural history is represented, the process and determinants of disease acquisition, and how the putative intervention could affect the system.

* Please provide a complete list of model parameters, including clear and precise descriptions of [the meaning of each parameter, together with the values or ranges for each, with justification or the primary source cited, and important caveats about the use of these values noted].

*Please provide a clear statement about how the model was fitted to the data [including goodness-of-fit measure, the numerical algorithm used, which parameter varied, constraints imposed on parameter values, and starting conditions].

* For uncertainty analyses, please state the sources of uncertainties quantified and not quantified [can include parameter, data, and model structure].

* Please provide sensitivity analyses to identify which parameter values are most important in the model. Uncertainty estimates seek to derive a range of credible results on the basis of an exploration of the range of reasonable parameter values. The choice of method should be presented and justified.

* Please discuss the scientific rationale for this choice of model structure and identify points where this choice could influence conclusions drawn. Please also describe the strength of the scientific basis underlying the key model assumptions.

Comments from the reviewers:

Reviewer #1: a) Could the authors elaborate on the 'simulated dynamic contact network' method used in the study. It would be beneficial for readers if this explanation is included either in the main manuscript or as an expanded section in the supplementary materials.

b) The manuscript mentions the use of a log-normal distribution for the duration individuals remain colonized and a normal distribution for the scheduling of swabbing days. Could the authors provide a detailed rationale for choosing these specific distributions? 

c) Regarding the equation for colonization duration that averages the days between the first and last positive swabs and the days around the last negative swab: the rationale for considering the entire period as colonization duration without division and the justification for halving the duration surrounding negative swabs warrant further clarification. It would be helpful if the authors could explain the underlying assumptions and their applicability more comprehensively.

d) Does the model incorporate probabilistic elements, including distribution-based variations for all parameters, or does it solely rely on deterministic simulations? Clarification on the use of probabilistic methods versus deterministic assumptions in parameter modelling would provide valuable insight into the model's complexity and its ability to simulate real-world variability.

e) The manuscript briefly discusses the impact of interventions such as re-allocation, contact precautions, and vaccination on model parameters, particularly highlighting a 6-fold reduction in transmission probability with vaccination implying an efficacy of 83%. Could the authors provide more extensive details on how these intervention effects were quantified and incorporated into the model? Specifically, the method for integrating the vaccination rate into the base-case scenario and its influence on model outcomes would be crucial for understanding the assumptions and implications of these interventions.

f) The manuscript would benefit from a more detailed presentation of the model validation and verification processes. While Figure 1 illustrates reproduced interactions, further elaboration on the methods and criteria used for model validation and verification would enhance the credibility and reliability of the model's findings.

Reviewer #2: Overall, I found this paper to be an enjoyable read, and an interesting one. I also commend the authors for making their code available. I do have a few potential areas where I believe the manuscript could use improvement:

1) The figures are very difficult to interpret in greyscale. More importantly, Figure 2 is nearly impossible to discern for several types of color blindness (using the CVSimulator app). I would suggest the authors consider one of several color blindness friendly pallets available in R.

2) The supplement does a good job describing the IBM, which is always a challenge. It would be nice if said description could be tailored to follow the MInD Framework, in hopes of making some progress in standardizing descriptions. https://academic.oup.com/cid/article/71/9/2527/5802665. I believe the supplement is not too far off from this already, and would represent relatively minimal effort.

3) The "reallocation" intervention is somewhat opaque. The authors discuss how *allocation* works in the methods, and one can, I think, reverse engineer from there, but I'm not positive of what I'm looking at from both an algorithmic level and how it would translate into clinical policy. It might also be good to place your results in the context (cohere with or disagree with) of Mietchen et al.'s paper (https://journals.plos.org/ploscompbiol/article?id=10.1371/journal.pcbi.1010352) looking at a compartmental model of staff assignment and MRSA in ICUs.

4) It would be good if the authors engaged with the likelihood of a colonization blocking vaccine vs. one that reduces onward transmission or other vaccine mechanisms, especially in light of said vaccine being largely hypothetical at this point.

Reviewer #3: Thank you for giving me the opportunity to review this interesting manuscript. Leclerc and colleagues take advantage of the i-Bird study to parametrize an agent-based hospital model with the aim to find the best interventions to counteract the spread of MRSA in long-term care facilities. While the manuscript tackles an important and critical issue in hospital hygiene, it would in my view benefit a lot by pushing it from a modelling exercise to a clinically usable source of evidence.

I have no major issue with any technical part of the modelling study, but have the feeling that clinicians and hospital hygiene experts were not consulted well enough. The selection of potential interventions and the way they are reported follows technical aspects, and the different model components (starting with manipulation of agent-agent contacts and the resulting contact networks, going through manipulation of transmission probabilities, and ending with increasing the efficiency of transmission probability manipulations through network-based identification of the best targets) are well known and also a bit trivial. The study needs to shine by bringing this together with the real-world paramerization of the model and subject-matter knowledge of what is feasible.

Since there is currently no recommended vaccine against MRSA carriage or infection, I would argue that this can be shifted to the supplement. All parameters chosen for the potential vaccine are hypothetical anyway. It is a nice visualisation of what happens in a model when you manipulate something - but in my opinion this has currently no real-world correlate and thus also no benefit.

The assumptions made for the effect of interventions on tranmission probability are broad - and of course affect the hierarchy of derived interventions. I would like the authors to decide - based on literature- what they assume to be the most appropriae parameter setting in each scenario - so that ranking by effectiveness and cost-effectiveness is posible. Something I think the manuscript needs; and I would like to see.

While the platform designed can of course be adapted to other MDR bacteria, I think the way this is discussed in the manuscript is too optimistic. Most MDR bacteria have a relevant transmission component via fomites which is not covered in i-Bird and thus undermines the USP of this manuscript - the real-world parametrization. This would not be possible for VRE or MDR-Enterobacteriacaeae.

Reviewer #4: The authors developed a detailed individual-based model to simulate MRSA transmission dynamics in a French long-term care facility. The study investigates at the impact of various interventions on MRSA colonization, differentiating between different types of interventions and different target populations. The authors created a contact network based on data collected from RFID sensor data recording close-proximity interactions from a previous study and were able to identify individuals that were influential in the contact network both in terms of frequency and the duration of the contact. The manuscript is generally well written and the dataset and the model itself are very detailed and as such interesting. However, I have concerns about the novelty of the results and relevance for public health. Most of the results in this study have been reported in the literature and so I would like to see more elaboration on how this study adds to known results. In addition, the authors use data from a single LTCF over a time period of only three months. This limits the generalizability of the results but also, it makes it difficult to fit the model to data over such a short time period. In particular, stochasticity of the data is hard to distinguish from 

In addition, I appreciate that several scenarios and sensitivity analyses were performed. However, the parameters for the scenarios seem to be arbitrarily picked and there is little explanation of how these parameters would translate to real-world scenarios in healthcare or whether there is existing evidence from the literature. The probably most surprising result was that the supercontactor to target differed between hospital staff and patients but the authors do not elaborate on these results nor provide further hypotheses for explanation. Generally, the Discussion section of the manuscript is more of a summary and repetition of the results from the study but I would like to see more of a "discussion" (I'll explain more in the line-by-line comments below): 

Line-by-line comments: 

Figure 1: 1b) shows the fit of the model to the weekly MRSA colonization data. I think this figure illustrates that over a short time period like this, there are a lot of stochastic effects even in the data (see the facet on Staff, second week of August). The prediction intervals of the model are very wide, sometimes ranging from 0 to 0.1. It's not clear to me that the model captures the dynamics well as stated by the authors. What are the measures by which the authors make that assessment? It seems that model fit was assessed only visually. To validate the model it would be useful to identify a measure by which model fit can be assessed more quantitatively (e.g. residual analysis or calibration plots) and provide these in the supplementary material. 

The numbering of the scenarios in all figures (e.g. Figure 2) seem unintuitive and random to me. Could the authors add more explanation on how these were numbered and why they are not presented in consecutive order? 

Figure 5 and 6: A 6-fold reduction in transmission probability in the main analysis seems quite high. It seems more likely that vaccination would have a smaller effect and maybe not even prevent acquisition of colonization but maybe only progression to clinical infection. What is the justification for using the 6-fold reduction in the main analysis? 

Lines 342-344: Would it be possible to add more explanation to this? 

Line 444: While the contact network data is very detailed and a valuable source, it is from 2009. Could the authors comment on changes in these facilities and how it could affect the network and consequently their results? 

Lines 502-508: The Reallocation intervention sounds similar to staff cohorting which I think is a more common used term. Did I understand that correctly? If yes, it could be helpful to mention that in the Methods section. 

Lines 509-522: The contact precaution scenarios assume reductions in transmission probability by 2,4,6,8,10 fold. What is the rationale behind these values? I think to see the application to public health, it would be useful to find literature on the impact of contact precautions (and their reinforcements)

---

## [Decision Letter · Decision Letter 2]

4 Jun 2024

Dear Dr. Leclerc,

Thank you very much for re-submitting your manuscript "Using contact network dynamics to implement efficient interventions against pathogen spread in hospital settings" (PMEDICINE-D-24-00101R2) for review by PLOS Medicine.

I have discussed the paper with my colleagues and the academic editor and it was also seen again by 2 reviewers. I am pleased to say that provided the remaining editorial and production issues are dealt with we are planning to accept the paper for publication in the journal.

[LINK]

If you have any questions in the meantime, please contact me at pdodd@plos.org or the journal staff on plosmedicine@plos.org.  

We look forward to receiving the revised manuscript by Jun 11 2024 11:59PM.   

Kind regards,

Pippa

Philippa Dodd, MBBS MRCP PhD

PLOS Medicine

plosmedicine.org

pdodd@plos.org

Requests from Editors:

COMMENTS FROM THE ACADEMIC EDITOR:

I have read through the revised manuscript and response to reviewers' comments. Overall, I think the authors have done well with their responses, and the revised manuscript is strengthened. I think the conclusions are now appropriately caveated regarding the hypothetical nature of the interventions.

COMMENTS FROM THE EDITORS:

GENERAL

Thank you for your detailed responses to previous editor and reviewer comments. Please see below for further comments which we require that you address prior to publication.

Specifically referring to the methods and results sections, please reformat and organize your manuscript according to PLOS Medicine’s formatting requirements. Specific comments are detailed below and further guidance can be found here https://journals.plos.org/plosmedicine/s/submission-guidelines#loc-manuscript-organization

Please note that many of the editorial requests detailed below pertain to specific formatting and content requirements. Some may have already been incorporated into the manuscript and some may not apply but please review the complete list of items and ensure that each item is included as necessary.

TITLE

Please revise your title according to PLOS Medicine's style. Your title must be nondeclarative and not a question. It should begin with main concept if possible. "Effect of" should be used only if causality can be inferred, i.e., for an RCT. Please place the study design ("A randomized controlled trial," "A retrospective study," "A modelling study," etc.) in the subtitle (i.e., after a colon).

DATA AVAILABILITY

Thank you for agreeing to make your code available which we appreciate. The Data Availability Statement (DAS) requires revision. As well as for the code, for each data source used in your study: 

STATISTICAL REPORTING

Throughout, including tables and figures, please quantify the main results with 95% CIs and p values.

When reporting p values please report as <0.001 and where higher as p=0.002, for example. If not reporting p values, for the purpose of transparent data reporting, please clearly state the reasons why not. When reporting 95% CIs please separate upper and lower bounds with commas instead of hyphens as the latter can be confused with reporting of negative values.

Please include the actual amounts and/or absolute risk(s) of relevant outcomes (including NNT or NNH where appropriate), not just relative risks or correlation coefficients. (example for absolute risks: PMID: 28399126).

LANGUAGE

Suggest caution with the language used to describe the different patient populations you study/simulate as ‘labels’ may cause offence. Please see below (methods and results) for more specific advice and please amend throughout all sub-sections of the manuscript and supporting files as necessary.

ABSTRACT

Abstract Background: Please provide context of why the study is important. The final sentence should clearly state the study question.

Abstract Methods and Findings:

Please ensure that all numbers presented in the abstract are present and identical to numbers presented in the main manuscript text.

Please include the study design, population and setting, number of participants, years during which the study took place, length of follow up, and main outcome measures.

Please quantify the main results with 95% CIs and p values as detailed above. If not, to help facilitate transparent data reporting, please clearly state the reasons why not.

Please include any important dependent variables that are adjusted for in the analyses.

Please include the actual amounts and/or absolute risk(s) of relevant outcomes (including NNT or NNH where appropriate), not just relative risks or correlation coefficients. (example for absolute risks: PMID: 28399126). 

In the last sentence of the Abstract Methods and Findings section, please describe the main limitation(s) of the study's methodology.

AUTHOR SUMMARY

At this stage, we ask that you include a short, non-technical Author Summary of your research to make findings accessible to a wide audience that includes both scientists and non-scientists. The authors summary should consist of 2-3 succinct bullet points under each of the following headings:

• Why Was This Study Done? Authors should reflect on what was known about the topic before the research was published and why the research was needed.

• What Did the Researchers Do and Find? Authors should briefly describe the study design that was used and the study’s major findings. Do include the headline numbers from the study, such as the sample size and key findings. 

• What Do These Findings Mean? Authors should reflect on the new knowledge generated by the research and the implications for practice, research, policy, or public health. Authors should also consider how the interpretation of the study’s findings may be affected by the study limitations. In the final bullet point of ‘What Do These Findings Mean?’, please describe the main limitations of the study in non-technical language.

The Author Summary should immediately follow the Abstract in your revised manuscript. This text is subject to editorial change and should be distinct from the scientific abstract. Please see our author guidelines for more information: https://journals.plos.org/plosmedicine/s/revising-your-manuscript#loc-author-summary

INTRODUCTION

Please ensure that you address past research and explain the need for and potential importance of your study. Indicate whether your study is novel and how you determined that. If there has been a systematic review of the evidence related to your study (or you have conducted one), please refer to and reference that review and indicate whether it supports the need for your study.

METHODS and RESULTS

As above, thank you for the extensive amendments you have made which have substantially improved clarity. 

Please move the ‘Methods’ sub-section to precede the ‘Results’ sub-section.

At times, the methodology is repeated and intertwined with the results – see lines 109 onwards as compared to lines 568 onwards, for example. The two subsections should exist as separate sub-sections (the methods preceding the results). Please amend throughout and to avoid repetition as much as possible.

Line 109 – suggest ‘neurology’.

Line 109 onwards – as above, suggest caution when describing patient populations. Please refrain from referring to ‘neurologic…PVS…patients’, for example. 

Line 110 – suggest ‘elderly care’.

Line 110 – suggest, ‘In addition, in-patients also included those in persistent vegetative state (PVS) and those in post operative recovery following general surgical and orthopaedic procedures.’ Or similar.

Line 113 – regarding the LCTF itself – is it designed for complex patients/procedures who/which are likely to require a long in-patient stay for either treatment or recovery? Additional nuanced details would be helpful for further context and to enable the reader to better appreciate transferability across settings.

Please present numerators and denominators used to derive percentages.

Please provide the actual numbers of events for the outcomes, not just summary statistics or relative estimates.

As above, where CIs are reported please also report p values. When a p value is given, please also include the test used to determine it.

TABLES and FIGURES

Please ensure that each table/figure (including those in the supporting files) is affiliated to an appropriate title/caption/footnote which clearly descries the content without the need to refer to the text.

Please ensure that all abbreviations (including those used in statistical reporting) are clearly defined in the footnote(s).

Please indicate the meaning of all dots, lines and bars in the caption/footnote for the reader.

Figure 4 – please change the label ‘Geriatry’ on the axis to ‘Elderly care’.

We thank you for using a color palate suitable to those with color blindness.

DISCUSSION

Please ensure that you present and organize the Discussion as follows: a short, clear summary of the article's findings; what the study adds to existing research and where and why the results may differ from previous research; strengths and limitations of the study; implications and next steps for research, clinical practice, and/or public policy; one-paragraph conclusion.

REFERENCES

For in-text reference callouts please place citations in square parentheses separate by commas. For example, [1,3,6] or [1-3]. Please check and amend throughout all sub-sections of the manuscript and supporting files.

In the bibliography, please ensure that you list up to but no more than 6 author names followed by et al.

For all web references please ensure you include an, ‘Accessed [date].’

Journal name abbreviations should be those listed in the National Center for Biotechnology Information (NCBI) databases.

SUPPORTING INFORMATION

Please ensure to apply all guidance detailed above to the supporting information files.

Please cite your Supporting Information as outlined here: https://journals.plos.org/plosmedicine/s/supporting-information

In the published article, supporting information files are accessed only through a hyperlink attached to the captions. For this reason, you must list captions at the end of your manuscript file. You may include a caption within the supporting information file itself, as long as that caption is also provided in the manuscript file. Do not submit a separate caption file.

SOCIAL MEDIA

To help us extend the reach of your research, if not already done so, please detail any X (formerly Twitter) handles you wish to be included when we tweet this paper (including your own, your coauthors’, your institution, funder, or lab) in the manuscript submission form when you re-submit the manuscript.

Comments from Reviewers:

Reviewer #4: The authors have considerably improved their manuscript and addressed most of the points I raised adequately. I specifically commend the addition of figures in the supplement to visualize the differences between the interventions. I have a couple of more questions/comments: 

Figure 1: Does the legend from 1a) also apply to 1b) (that's what it seems like). Would it be possible to place the legend such that it is clear that it's for both figures?

Figure S8: This figure is helpful to compare the effects of the different interventions for different assumptions of the vaccine efficacy. However, it took me a while to understand the figure. Would it be possible to extend

---

## [Editor Report · Decision Letter 3]

24 Jun 2024

Dear Dr Leclerc, 

On behalf of my colleagues and the Academic Editor, Professor Peter MacPherson, I am pleased to inform you that we have agreed to publish your manuscript "Using contact network dynamics to implement efficient interventions against pathogen spread in hospital settings: A modelling study" (PMEDICINE-D-24-00101R3) in PLOS Medicine.

Prior to publication, at the time you make your formatting changes as detailed below, please also make the following amendments:

1) Data Availability Statement: 

As you leveraged data from the i-Bird study please ensure that the relevant details for readers to access these data are also included.

2) Tables/figures – main manuscript: 

Please ensure the caption labels match the citations and the file names. The 1st figure should be labelled as ‘Fig 1’ (as opposed to Figure 1) and cited as such. File names should also match i.e., ‘Fig 1.tiff’ please amend as necessary. 

Please apply the same to tables as necessary.

3) Supporting Information: 

Please label tables as ‘S1 Table’ (so on).

Please label figures as ‘S1 Fig’ (and so on).

Any additional documents (protocols/analysis plans etc.) can be labelled as ‘S1 Protocol’, for example.

Please cite items as exactly as labelled.

Further guidance can be found here https://journals.plos.org/plosmedicine/s/supporting-information

PRESS

Thank you again for submitting to PLOS Medicine. It has been a pleasure handling your manuscript, we look forward to publishing your paper. 

Kind regards, 

Pippa

Philippa Dodd, MBBS MRCP PhD 

Senior Editor 

PLOS Medicine

pdodd@plos.org